# Antibiotic resistance, pathotypes, and pathogen-host interactions in *Escherichia coli* from hospital wastewater in Bulawayo, Zimbabwe

**Joshua Mbanga** [1]*, **Nokukhanya P. Kodzai**[1], **Wilhem F. Oosthuysen**[2]

**1** Department of Applied Biology and Biochemistry, National University of Science and Technology, Bulawayo, Zimbabwe, **2** Nandrea Health Products, Oudtshoorn, Western Cape, South Africa

* joshua.mbanga@nust.ac.zw

**Data Availability Statement:** All relevant data are within the paper and its Supporting Information files.

## Abstract

This study aimed to characterise *E. coli* strains isolated from hospital wastewater effluent in Bulawayo, Zimbabwe, using both molecular and cytological approaches. Wastewater samples were aseptically collected from the sewerage mains of a major public referral hospital in Bulawayo province weekly for one month. A total of 94 isolates were isolated and confirmed as *E. coli* through biotyping and PCR targeting the *uidA* housekeeping gene. A total of 7 genes (*eagg*, *eaeA*, *stx*, *flicH7*, *ipaH*, *lt*, and *st* genes) coding for virulence in diarrheagenic *E. coli* were targeted. Antibiotic susceptibility of *E. coli* was determined against a panel of 12 antibiotics through the disk diffusion assay. The infectivity status of the observed pathotypes was investigated using HeLa cells through adherence, invasion, and intracellular assay. None of the 94 isolates tested positive for the *ipaH* and *flicH7*genes. However, 48 (53.3%) isolates were enterotoxigenic *E. coli* (ETEC) (*lt* gene positive), 2 (2.13%) isolates were enteroaggregative *E. coli* (EAEC) (*eagg* gene), and 1 (1.06%) isolate was enterohaemorrhagic *E. coli* (EHEC) (*stx* and *eaeA*). A high level of sensitivity was observed in *E. coli* against ertapenem (98.9%), and Azithromycin (75.5%). The highest resistance was against ampicillin (92.6%) and sulphamethoxazole—trimethoprim (90.4%). Seventy-nine (84%) *E. coli* isolates exhibited multidrug resistance. The infectivity study results indicated that environmentally isolated pathotypes were as infective as the clinically isolated pathotypes for all three parameters. No adherent cells were observed using ETEC, and no cells were observed in the intracellular survival assay using EAEC. This study revealed that hospital wastewater is a hotspot for pathogenic *E. coli* and that the environmentally isolated pathotypes maintained their ability to colonise and infect mammalian cells.

## Introduction

Antibiotic resistance (AR) is an unprecedented global public health concern as it has implications for the successful treatment of human and animal infections [1, 2]. Hospital wastewater

**Funding:** This work is based on the research supported by the National University of Science and Technology (NUST) Research Board grant # RB17/43 and the kind support of the Vaccine-Preventable Diseases Respiratory and Meningeal Pathogens Research Unit (RMPRU), South Africa. The funders had no role in study design, data collection, and analysis, decision to publish, or preparation of the manuscript.

**Competing interests:** The authors have declared that no competing interests exist.

is recognised to be among the major reservoirs of antibiotic-resistant bacteria (ARB) and antibiotic-resistance genes (ARGs), as it possesses higher levels of antibiotics, disinfectants, and patient excrement. The exposure of these pathogens to a wide range of biocides and antibiotics may result in pathogenic bacteria that harbour resistant genes to biocides, heavy metals, and antibiotics [3]. The hospital wastewater environment is a potential source of multidrug resistant pathogenic bacteria, including methicillin-resistant *S. aureus* (MRSA), vancomycin-resistant *Enterococcus* (VRE), and extended-spectrum beta-lactamase (ESBL) producing Gram-negative bacteria. Hospital wastewater can provide a suitable environment for the interaction between different bacteria, facilitating the exchange of ARGs between clinical and environmental bacterial strains [4]. This can result in significant public health and epidemiological consequences that extend beyond hospitals into neighbouring communities [5, 6].

While most strains of *E. coli* are commensal, virulent strains that can cause a variety of intra-intestinal diseases, such as diarrhoea, and extra-intestinal diseases, such as urinary tract infections, exist [7, 8]. At present, the enteric *E. coli* pathotypes, also known as Diarrhoeagenic *E. coli* (DEC), are divided into six groups based on their serological virulence- characteristics and other phenotypic traits. The six groups are enterotoxigenic *E. coli* (ETEC), enteropathogenic *E. coli* (EPEC), enterohaemorrhagic *E. coli* (EHEC), or Verotoxin or Shiga toxin-producing *E. coli* (VTEC/STEC) enteroinvasive *E. coli* (EIEC), diffusely adherent *E. coli* (DAEC), and enteroaggregative *E. coli* (EAEC) [9, 10].

Although using molecular methods to examine and characterise *E. coli* pathotypes and other bacteria is essential, these methods provide only a partial understanding of these bacterial strains. Cytological approaches have been employed to enhance the understanding of these pathotypes and their behaviour when interacting with human cells [11, 12]. The first step in understanding the effect of pathogenic bacteria on the human body is to examine their impact on human cell lines, and this is done through adherence, invasion, and intracellular survival assays, also known as infectivity studies [13]. These assays may be used to categorise different *E. coli* pathotypes based on their ability to adhere, invade and survive inside human cells. For example, enteroaggregative and diffusely adhering *E. coli* may be defined by their degree of adherence to HeLa cells and their cytotoxic activity *in vitro* [7, 14].

A lot of studies have been done on *E. coli* characterisation based on cytological and molecular studies, with more studies done in Europe and Asia [15, 16]. Therefore, this study focused on the identification of DEC strains in hospital wastewater and investigated their infectivity status using a human cell line.

## Materials and methods

### Ethical approval

Ethical approval for this study was obtained from the NUST Institution Research Board (NUST/IRB/202/34).

### Study site and sample collection

Wastewater samples were aseptically collected from sewerage mains located at the western wing of a major public referral hospital in Bulawayo province, Zimbabwe. The hospital is one of two principal referral centres in the Southern part of Zimbabwe and has a bed capacity of 650. The sampled sewerage mains receives liquid waste from the Casualty department, Burns ward, Intensive Care Unit (ICU), Female ward, and Children's ward. Sample collection was done once a week for one month from February to March 2020. Sample collection was done using grab water sampling into a 500ml sterile plastic container. A total of 4 wastewater samples were collected in duplicate and transported in a cooler box on ice to the NUST

Microbiology laboratory for analysis. The duplicate samples were pooled before analysis to make a composite sample. Some images of the sampling site are shown in S1 Fig. All collected samples were processed within 2h of collection, and the remaining samples were stored at 4°C until all analysis was complete.

## Isolation of *E. coli*

Ten-fold serial dilutions of the collected samples were carried out by diluting 1ml of the collected samples in 9ml of autoclaved distilled water for dilutions $10^{-1}$ up to $10^{-5}$. After that, 100μl of the $10^{-1}$, $10^{-2}$, $10^{-3}$, $10^{-4}$, and $10^{-5}$ dilutions were cultured in duplicates using the spread plate method on Levine Eosin Methylene Blue (EMB) Agar (HiMedia Labs, India). Inoculated plates were incubated for 24 h at 37°C, thereafter, typically metallic green sheen colonies were sub-cultured as presumptive *E. coli* isolates. To subculture the isolates, 10 single colonies were randomly picked per sampling time point from the $10^{-3}$, $10^{-4}$, and $10^{-5}$ plates and triple streaked on Plate Count Agar plates, before incubation at 37°C for 24 hours. After 24 h, the Gram stain and the oxidase tests were done on the presumed *E. coli* isolates. Presumptive isolates were stored in 30% glycerol stocks at -70°C until further use.

## Molecular confirmation of *E. coli*

*E. coli* DNA was extracted using a standard heat lysis protocol [17]. To check for purity the crude DNA was run on a ethidium bromide stained 1% agarose gel (Sigma-Aldrich, St Louis, USA) with a 100bp DNA ladder (Thermo Scientific, Johannesburg, SA) in 1x TBE buffer for 1hr at 100V and then viewed using a Uvipro-Silver Gel Documentation System (Uvitec, UK). The concentration of DNA was estimated by comparing the band light intensity to the band intensity on the 100 bp ladder on the Uvipro-Silver Gel Documentation System. Crude DNA between 5 – 10ng/μl was used for PCR reactions. The *uid*A (β-D glucuronidase) gene was used to confirm *E. coli* using conventional PCR [18]. The PCR reaction consisted of a total reaction mixture of 10uL with 5 μL of 2x Dreamtaq master mix (New England Biolabs, UK), 0.16 μL (0.4 μM) forward primer (AAAACGGCAAGAAAAAGCAG), 0.16 μL (0.4 μM) reverse primer (ACGCGTGGTTAACAGTCTTGCC) (Inqaba Biotech, Pretoria, SA), 2.6 μL nuclease-free water (New England Biolabs, UK) and 2 μL of DNA template was prepared and run on a MiniAmp™ Plus Thermal Cycler (Applied Biosystems, Thermo Fisher Scientific, USA). The negative control comprised nuclease-free water in place of template DNA. The PCR profile was as follows: Initial denaturation at 95°C for 3 minutes, 30 Cycles {denaturation at 95°C for 30 seconds, annealing temperatures at 50°C for 30 seconds, extension at 72°C for 1 minute} and then the final extension at 72°C for 5 minutes. *E. coli* ATCC® 25922 was used as a positive control.

## Pathotyping of *E. coli*

*E. coli* was pathotyped via conventional PCR. The pathotypes assayed for included EHEC, EPEC, EIEC, EAEC, and ETEC. *E. coli* DSM reference strains (DSM 8695, DSM 10 973, DSM 10 974, DSM 9025, O157H7) were included in the assays as internal positive controls for the targeted *E. coli* pathotypes. The virulence gene primers and reference strains used in this study are shown in S1 Table. The PCR comprised of a reaction mixture of 10uL consisting of 5 μL of 2x Dreamtaq master mix (New England Biolabs, UK), 0.16 μL (0.4 μM) forward primer, 0.16 μL (0.4 μM) reverse primer (Inqaba Biotech, Pretoria, SA), 2.6 μL nuclease-free water (New England Biolabs, UK) and 2 μL of DNA template was prepared and run on a MiniAmp™ Plus Thermal Cycler (Applied Biosystems, Thermo Fisher Scientific, USA). The negative control comprised nuclease-free water in place of template DNA. The PCR profile was as

follows: Initial denaturation at 95˚C for 3 minutes, 30 Cycles {denaturation at 95˚C for 30 seconds, the following annealing temperatures were used; 42˚C for *stx*, 48˚C for *flicH* 7 and lt, 50˚C for *ipaH* genes, 53.5˚C for *eaeA*, 55˚C for *eagg*, and 63˚C for *st*, for 30–45 seconds; extension at 72˚C for 1 minute} and then the final extension at 72˚C for 5 minutes. The PCR products were all electrophoresed in a 1% agarose gel containing 0.5mg/l ethidium bromide (Sigma-Aldrich, St. Louis, USA) for 50 min at 100V in 1x TBE buffer. Amplicon sizes were then visualized using a UV transilluminator (Uvitec, Cambridge, UK) and run alongside a Invitrogen 1kb plus DNA ladder (Thermo Scientific, Johannesburg, SA) as a standard molecular weight marker.

**Antibiotic susceptibility testing of *E. coli*.**    Antibiotic susceptibility of *E. coli* was determined against a panel of 12 antibiotics through the disk diffusion assay. A 0.5% McFarland's standard was used to standardize all innocula prior to carrying out the antibiotic susceptibility test. The commercial antibiotic discs included ampicillin (AMP, 10μg), azithromycin (AZM, 15μg), amoxicillin-clavulanic acid (AMC, 30μg), cefotaxime (CTX, 30μg), cefepime (FEP, 10μg), ceftriaxone (CRO, 30μg), cefixime (CFM, 5μg), ciprofloxacin (CIP, 5μg), ertapenem (ETP, 10μg), tetracycline (TET, 30μg), trimethoprim-sulfamethoxazole (SXT, 1.25μg/23.75μg), and chloramphenicol (CHL, 30μg) (MAST group, United Kingdom). Antibiotic susceptibility testing and interpretation of results were done following the Clinical and Laboratory Standards Institute (CLSI) guidelines and interpretive charts [19, 20]. *E. coli* ATCC 25922 was used as a reference strain. Multidrug resistance (MDR) was defined as resistance to one or more antibiotics in three or more antibiotic classes [21].

## Adherence and invasion assay

Three isolates belonging to the ETEC (BHE16), EAEC (BHE59), and EHEC (BHE94) pathotypes were used for the assays. The environmental isolates were randomly chosen except for BHE94, which was the only EHEC isolate. The characteristics of the chosen isolates are shown in Table 1. The positive controls of the corresponding pathotypes were also used for comparison. The assays were done according to Sinha et al. [22] with modifications. Briefly, HeLa cells were seeded at $1.5 \times 10^6$ cells/well in a 24-well plate and cultured at 37˚C/5% in an EC 160 $CO_2$ humidified incubator (Nüve, Turkey) for 24h to allow for the formation of a monolayer. All adherence and invasion experiments were performed in duplicate per strain, time point, and multiplicity of infection (MOI). Three independent experiments were performed. A volume of 1 ml invasion medium (DMEM with bacteria at the MOI of 10) was added to each well and incubated at 37˚C/5% $CO_2$ in a humidified incubator for two-time points. Adherence and invasion were investigated at the 1h and 2h time points after the addition of the invasion medium to the cellular monolayers. Following incubation, infected cellular monolayers were washed 3× with DMEM, trypsinized with 200μl trypsin, and incubated for 10 min at room temperature. Trypsinized cells were neutralised by 600μl DMEM/10% FCS, removed from

**Table 1. Antibiotic susceptibility and pathotypes of three *E. coli* isolates used in the infectivity study.**

| Isolate ID | Pathotype | ETP | SXT | CTX | CRO | FEP | CFM | TET | CIP | AMP | AZM | CHL | AMC | Resistance pattern |
|---|---|---|---|---|---|---|---|---|---|---|---|---|---|---|
| BHE16 | ETEC | S | R | R | R | I | R | I | I | R | S | R | R | SXT-CTX-CRO-CFM-AMP-CHL-AMC |
| BHE59 | EAEC | S | R | S | I | S | S | S | S | R | S | S | S | SXT-AMP |
| BHE94 | EHEC | S | R | R | I | I | S | S | S | R | R | S | R | SXT-CTX-AMP-AZM-AMC |

**KEY**: S- susceptible; I- intermediate; R–Resistant; ampicillin (AMP), Azithromycin (AZM), amoxicillin-clavulanic acid (AMC), cefepime (FEP), cefotaxime (CTX), ceftriaxone (CRO), cefixime (CFM), chloramphenicol (CHL), ertapenem (ETP), trimethoprim-sulfamethoxazole (SXT) ETEC–Enterotoxigenic *E. coli*; EAEC–Enteroaggregative *E. coli*; EHEC–Enterohaemorrhagic *E. coli*.

each well, and collected in a 1.5ml tube. The cellular suspension was centrifuged at 800rpm for 10 min to separate bacterial cells from human cells, the supernatant was then transferred to a new 1.5ml tube for bacterial adherence, and the cellular pellet was used for bacterial invasion.

To determine bacterial adherence, the supernatant was collected into a new tube and centrifuged at 12 000 rpm for 10 min. The supernatant was aspirated, and the bacterial pellet was re-suspended in 200 μl PBS, transferred to a 96-well plate, and serially diluted. A total of 20 μl of the $10^{-3}$–$10^{-6}$ dilutions were plated on LB agar plates and incubated at 37˚C for 24 hours. Following the incubation, the plates were left at room temperature for 1h to dry before counting the colony-forming units (CFUs).

To determine bacterial invasion, the bacterial pellet from the original tube was re-suspended in 200 μl 1% Triton-X and incubated for 10 min at 37˚C. Following the incubation, 20 μl was serially diluted as described for bacterial adherence. The same dilutions $10^{-3}$–$10^{-6}$ were plated on LB agar and incubated before counting the CFUs.

### Intracellular survival

Utilising the same time points used for bacterial adherence and invasion, the invasion medium was removed, and the cellular monolayer was washed 3× with DMEM. A total of 1ml DMEM containing PENSTREP 20/25 (100μg/μl) was added to each well, and the plates were incubated for an additional 2h at 37˚C/5% $CO_2$ in a humidified incubator to lyse any remaining adherent and invasive organism and obtain those which survived intracellularly. Following the 2h incubation, the medium was removed, and the cellular monolayer was washed 3× with DMEM. A total of 200μl 1% Triton-X was added to each well and incubated for 10 min at 37˚C/5% $CO_2$ in a humidified incubator to lyse the human cells and release any surviving intracellular organisms. Serial dilutions were prepared, and $10^{-1}$–$10^{-4}$ dilutions were plated on LB agar plates. LB plates were incubated as described, and bacterial CFUs were enumerated following incubation.

### Statistical analysis

Statistical analysis was done on Microsoft Excel 2018 and GraphPad Prism 8.4.3 software. GraphPad Prism was used to analyse the adherence, invasion, and survival assays by performing a two-way ANOVA at a 95% Confidence Interval (p = 0.05).

## Results

### Isolation and identification of *E. coli*

From 120 presumptive *E. coli*, a total of 94 (week 1, 25; week 2, 23; week 3, 30; week 4, 16) isolates were confirmed as *E. coli* using conventional PCR of the *uidA* gene.

### Antibiotic susceptibility of *E. coli*

The sensitivity of *E. coli* to the assayed antibiotics is presented in Table 2. For all tested isolates, a high level of sensitivity was observed against ETP 93(98.9%) and AZM 71(75.5%). Elevated resistance was observed against most of the tested antibiotics, with AMP 87(92.6%), SXT 85 (90.4%), and CTX 78(83%) having the highest resistance. Elevated levels of multidrug resistance were observed, with 79 (84%) *E. coli* isolates exhibiting resistance to antibiotics from at least three classes. A total of 24 MDR patterns were observed across all isolates, with the most common being pattern A (SXT-CTX-CRO-CFM-TET-AMP-CHL-AMC), which occurred in 24 isolates (Table 3). One isolate was resistant to 11 out of 12 antibiotics used in this study. Most *E. coli* belonging to the identified pathotypes were multidrug resistant (Table 4). A ETEC

**Table 2. Antibiotic sensitivity of confirmed *E. coli* isolates from hospital wastewater.**

| Antibiotic (µg) | Resistant no. (%) | Intermediate no. (%) | Susceptible no. (%) |
|---|---|---|---|
| Ertapenem (10) | 0(0) | 1(1.1) | 93(98.9) |
| Trimethoprim-Sulfamethoxazole (1.25/23.75) | 85(90.4) | 1(1.1) | 8(8.5) |
| Cefotaxime (30) | 78(83) | 6(6.4) | 10(10.6) |
| Ceftriaxone (30) | 75(79.8) | 10(10.6) | 9(9.6) |
| Cefepime (10) | 40(42.6) | 42(44.7) | 12(12.7) |
| Cefexime (5) | 73(77.7) | 0(0) | 21(22.3) |
| Tetracycline (30) | 69(73.4) | 5(5.4) | 20(21.2) |
| Ciprofloxacin (5) | 13(13.8) | 54(57.5) | 27(28.7) |
| Ampicillin (10) | 87(92.6) | 3(3.2) | 4(4.2) |
| Amoxicillin-clavulanate (30) | 73(77.7) | 1(1.1) | 20(21.2) |
| Azithromycin (15) | 23(24.5) | 0(0) | 71(75.5) |
| Chloramphenicol (30) | 64(68.1) | 1(1.1) | 29(30.8) |

isolate (BHE72) was susceptible to all tested antibiotics, whilst another isolate (BHE73) was only resistant to TS. The occurrence of multidrug resistant *E. coli* pathotypes in the hospital wastewater suggests that these may be widespread in the hospital environment.

**Table 3. Antibiotic resistance patterns of *E. coli* isolates from hospital wastewater.**

| Pattern | Number of isolates | Resistance pattern |
|---|---|---|
| A | 24 | SXT,CTX,CRO,CFM,TET,AMP,CHL,AMC |
| B | 13 | SXT,CTX,CRO,FEP,CFM, TET, AMP,AZM, CHL, AMC |
| C | 10 | SXT,CTX,CRO, FEP,CFM, TET, AMP, CHL, AMC |
| D | 5 | SXT,CTX,CRO, FEP,CFM, TET,CIP, AMP, AZM, AMC |
| E | 4 | SXT,CTX,CRO,CFM, AMP, CHL, AMC |
| F | 2 | CTX,CRO,CFM, TET,CIP, AMP, CHL, AMC |
| G | 2 | SXT,CTX,CRO, FEP,CFM, TET,CIP, AMP, CHL, AMC |
| H | 2 | SXT,CTX,CRO, AMP |
| I | 2 | SXT,CTX, AMP |
| J | 1 | SXT,CTX,CRO,CFM, TET,CHL,AMC |
| K | 1 | CTX,CRO,CFM,TET, AMP,CHL, AMC |
| L | 1 | CFM, TET, AMC |
| M | 1 | CTX,CFM, TET,AMP,AMC |
| N | 1 | SXT,CTX,CRO,CFM, TET,CIP, AMP, CHL, AMC |
| O | 1 | CTX,CRO, FEP,CFM, TET, AMP, CHL, AMC |
| P | 1 | CTX,CRO, FEP,CFM, TET, AMP, AZM, CHL, AMC |
| Q | 1 | SXT,CTX,CRO,CFM, TET, AMP, AZM, CHL, AMC |
| R | 1 | SXT,CRO, FEP |
| S | 1 | SXT, CTX,CRO, FEP,CFM,TET, AMP, AZM, AMC |
| T | 1 | SXT,CTX,CRO, FEP,CFM,TET,CIP, AMP, AZM,CHL,AMC |
| U | 1 | SXT,CTX,CRO, FEP,CFM,TET,CIP, AMP,AMC |
| V | 1 | SXT,CTX,CRO, FEP,AMP |
| W | 1 | SXT, FEP, AMP |
| X | 1 | SXT,CTX, AMP, AZM, AMC |

**Key**: ampicillin (AMP), Azithromycin (AZM), amoxicillin-clavulanic acid (AMC), cefepime (FEP), cefotaxime (CTX), ceftriaxone (CRO), cefixime (CFM), chloramphenicol (CHL), ertapenem (ETP), trimethoprim-sulfamethoxazole (SXT), tetracycline (TET), ciprofloxacin (CIP).

**Table 4. Antibiotic resistance patterns of pathogenic *E. coli* pathotypes.**

| Isolate ID | Virulence gene present | Pathotype | Antibiogram |
|---|---|---|---|
| BHE1 | *lt* | ETEC | TS,CTX,CRO,CPM,CFM,T,AP,C,AUG |
| BHE2 | *lt* | ETEC | TS,CTX,CRO,CPM,CFM,T,AP,C,AUG |
| BHE3 | *lt* | ETEC | TS,CTX,CRO,CFM,T,AP,C,AUG |
| BHE4 | *eagg* | EAEC | TS,CTX,CRO,CFM,T,AP,C,AUG |
| BHE5 | *lt* | ETEC | TS,CTX,CRO,CFMT,AP,C,AUG |
| BHE6 | *lt* | ETEC | TS,CTX,CRO,CPM,CFM,T,AP,C,AUG |
| BHE7 | *lt* | ETEC | TS,CTX,CRO,CFM,T,AP,C,AUG |
| BHE8 | *lt* | ETEC | TS,CTX,CRO,CPM,CFM,T,AP,C,AUG |
| BHE9 | *lt* | ETEC | TS,CTX,CRO,CFM,T,AP,C,AUG |
| BHE10 | *lt* | ETEC | TS,CTX,CRO,CFM,T,AP,C,AUG |
| BHE11 | *lt* | ETEC | TS,CTX,CRO,CFM,T,AP,C,AUG |
| BHE12 | *lt* | ETEC | TS,CTX,CRO,CFM,T,AP,C,AUG |
| BHE13 | *lt* | ETEC | TS,CTX,CRO,CFM,T,AP,C,AUG |
| BHE14 | *lt* | ETEC | TS,CTX,CRO,CFM,T,AP,C,AUG |
| BHE15 | *lt* | ETEC | TS,CTX,CRO,CFM,T,AP,C,AUG |
| BHE16 | *lt* | ETEC | TS,CTX,CRO,CFM,AP,C,AUG |
| BHE18 | *lt* | ETEC | TS,CTX,CRO,CFM,AP,C,AUG |
| BHE19 | *lt* | ETEC | TS,CTX,CRO,CFM,T,AP,C,AUG |
| BHE20 | *lt* | ETEC | TS,CTX,CRO,CFM,AP,C,AUG |
| BHE21 | *lt* | ETEC | TS, CTX,CRO,CFM,T,C,AUG |
| BHE22 | *lt* | ETEC | TS, CTX,CRO,CFM,T,AP,C,AUG |
| BHE23 | *lt* | ETEC | CTX, CRO,CFM,T,AP,C,AUG |
| BHE24 | *lt* | ETEC | CFM, T,AUG |
| BHE25 | *lt* | ETEC | CTX, CFM,T,AP,AUG |
| BHE26 | *lt* | ETEC | CTX, CRO,CFM,T,CIP,AP,C,AUG |
| BHE27 | *lt* | ETEC | TS, CTX,CRO,CFM,T,AP,C,AUG |
| BHE28 | *lt* | ETEC | TS, CTX,CRO,CFM,T,AP,C,AUG |
| BHE29 | *lt* | ETEC | TS, CTX,CRO,CFM,T,CIP,AP,C,AUG |
| BHE30 | *lt* | ETEC | TS, CTX,CRO,CFM,T,AP,C,AUG |
| BHE31 | *lt* | ETEC | TS, CTX,CRO,CPM,CFM,T,AP,C,AUG |
| BHE33 | *lt* | ETEC | TS, CTX,CRO,CFM,T,AP,C,AUG |
| BHE36 | *lt* | ETEC | TS, CTX,CRO,CFM,T,AP,C,AUG |
| BHE46 | *lt* | ETEC | TS, CTX,CRO,CFM,T,AP,C,AUG |
| BHE48 | *lt* | ETEC | TS, CTX,CRO,CPM,CFM,T,AP,C,AUG |
| BHE50 | *lt* | ETEC | TS, CTX,CRO,CPM,CFM,T,AP,ATH,C,AUG |
| BHE56 | *lt* | ETEC | TS, CTX,CRO,CPM,CFM,T,CIP,AP,ATH,,AUG |
| BHE58 | *lt* | ETEC | TS, CTX,CRO,CPM,CFM,T,CIP,AP,ATH,,AUG |
| BHE59 | *eagg* | EAEC | TS, AP |
| BHE64 | *lt* | ETEC | TS,CTX,CRO,CPM,CFM,T,AP,ATH,C,AUG |
| BHE68 | *lt* | ETEC | TS,CTX,CRO,CPM,CFM,T,AP,ATH,C,AUG |
| BHE69 | *lt* | ETEC | TS,CTX,CRO,CPM,CFM,T,AP,ATH,C,AUG |
| BHE70 | *lt* | ETEC | TS,CTX,CRO,CPM,CFM,T,CIP,AP,ATH,,AUG |
| BHE72 | *lt* | ETEC | |
| BHE73 | *lt* | ETEC | TS |
| BHE76 | *lt* | ETEC | TS,CTX,CRO,CPM,CFM,T,AP,ATH,,AUG |
| BHE79 | *lt* | ETEC | TS,CTX,CRO,CPM,CFM,T,CIP,AP,AUG |

(*Continued*)

**Table 4.** (Continued)

| Isolate ID | Virulence gene present | Pathotype | Antibiogram |
|---|---|---|---|
| BHE94 | *eaeA, sxt* | EHEC | TS,CTX,AP,ATH,AUG |

**Key**: ampicillin (AMP), Azithromycin (AZM), amoxicillin-clavulanic acid (AMC), cefepime (FEP), cefotaxime (CTX), ceftriaxone (CRO), cefixime (CFM), chloramphenicol (CHL), ertapenem (ETP), trimethoprim-sulfamethoxazole (SXT), tetracycline (TET), ciprofloxacin (CIP): ETEC–Enterotoxigenic *E. coli*; EAEC–Enteroaggregative *E. coli*; EHEC–Enterohaemorrhagic *E. coli*.

## Distribution of *E. coli* pathotypes

*E. coli* pathotype distribution showed that ETEC was the most prevalent, with 48 isolates (51.1%), followed by EAEC with two isolates (2.1%) (Fig 1). The least observed pathotype was the EHEC, with 1 (1.1%) isolate. No isolates tested positive for EPEC and EIEC pathotypes. Weekly distribution showed a general increase in the number of pathotypes detected over one month, with the ETEC pathotype being consistently seen throughout (Fig 2).

## Adherence, invasion, and intracellular survival of *E. coli* pathotypes

The adherence, invasion, and survival of the three pathotypes observed in this study (ETEC, EAEC, and EHEC) were investigated using HeLa cells. The results were analysed using a two-way ANOVA at 95% Confidence Interval (p = 0.05). None of the EAEC strains survived inside the HeLa cells, and none of the ETEC adhered to the cells. The results generally showed no significant difference in the level of infectivity between the clinical control strain and the corresponding environmental isolate (Fig 3). However, there was a generally significantly higher level of infectivity after 2 hours of exposure. Only EHEC and EAEC adhered to Hela cells. There was no significant difference in the adherence of both pathotypes' clinical and environmental isolates at the two time points (Fig 3A). However, adherence was significantly higher after 2 hours for the EAEC, with a mean of 6.978 and 7.39 log CFU/ml for the clinical and environmental isolate respectively. All pathotypes were invasive (Fig 3B) with EAEC and ETEC

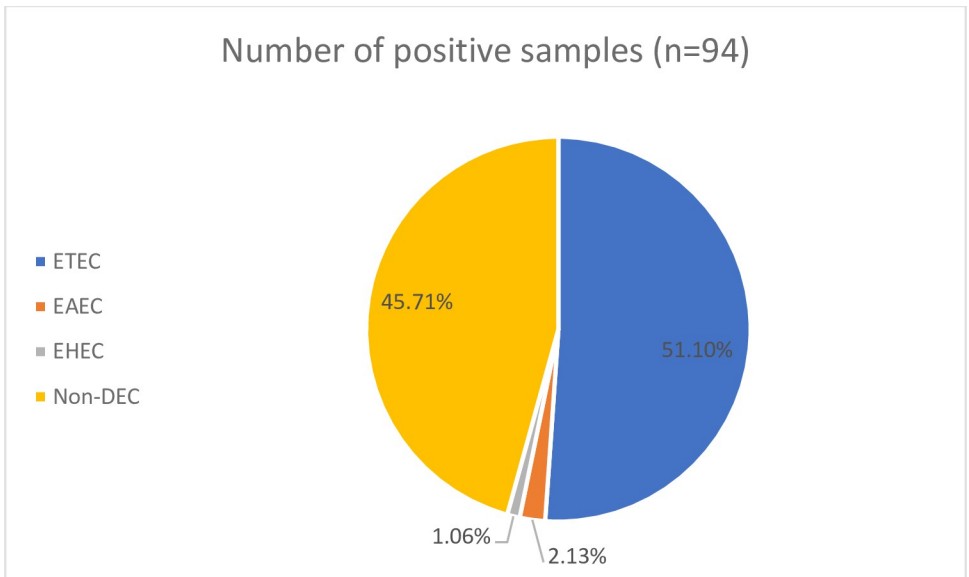

**Fig 1. Distribution of *E. coli* pathotypes isolated from hospital wastewater over one month.** Enteroinvasive E. coli and Enteropathogenic E. coli were not detected during the study. **Key:** ETEC–Enterotoxigenic *E. coli*; EAEC–Enteroaggregative *E. coli*; EHEC–Enterohaemorrhagic *E. coli*; DEC- diarrhoeagenic *E. coli*.

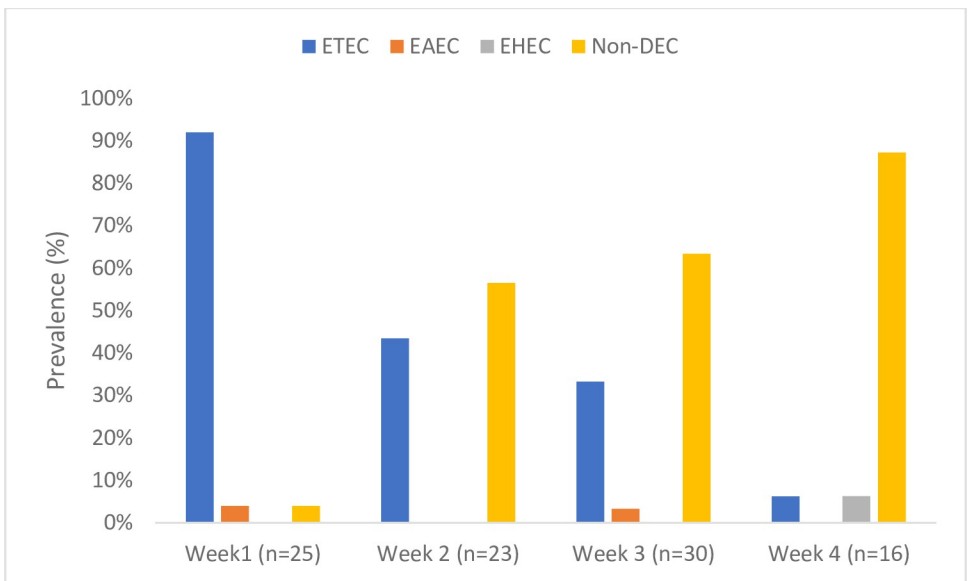

**Fig 2. Weekly distribution of *E. coli* pathotypes obtained from hospital effluent. Key:** ETEC–Enterotoxigenic *E. coli*; EAEC–Enteroaggregative *E. coli*; EHEC–Enterohaemorrhagic *E. coli*; DEC- diarrhoeagenic *E. coli*.

showing no significant difference between invasion by the experimental and clinical isolates at both time points. There, however, was a significant difference in invasion by the clinical and experimental EHEC isolates at both time points. Invasion of Hela cells increased significantly for the EHEC and EAEC after 2 hours (Fig 3B). Intracellular survival was observed for the EHEC and ETEC pathotypes (Fig 3C). There was no significant difference in intracellular survival of the ETEC clinical and environmental isolates. The clinical and environmental EHEC isolates were significantly different. Notably, only the clinical isolate increased significantly intracellularly after 2 hours.

## Discussion

### Antibiotic resistance profiles of *E. coli*

Most of the *E. coli* isolates had high susceptibility to ertapenem (98.9%) and azithromycin (73%) (Table 2). Ertapenem is a carbapenem and is not routinely used in Zimbabwe as it is considered a last-resort antibiotic that is reserved for serious infections. Azithromycin is a macrolide that can help treat a wide range of bacterial infections and has been shown to still be effective against endemic pathogens like *Salmonella* Typhi [23]. The *E. coli* isolates showed the highest resistance to AMP (92.6%), SXT (90.4%), and CTX (83%) (Table 2). The current findings were similar to those reported in other studies on untreated hospital wastewater from Ethiopia [24, 25]. Both studies reported elevated resistance to ampicillin amongst isolated *E. coli*. As expected, a high number of isolated *E. coli* 79 (84%) were multidrug resistant; this was much higher than that reported by King et al. [26], who isolated *Klebsiella* spp. in untreated effluent from an urban and rural hospital in KwaZulu-Natal province in South Africa, and reported that 23% (urban hospital) and 9% (rural hospital) of the isolates were MDR.

### Prevalence of diarrhoeagenic *E. coli* in hospital wastewater

Among the five DEC pathotypes profiled from the 94 confirmed *E. coli* isolates, the ETEC pathotype was the most prevalent 48 (51.1%). However, all isolates only had the *lt* gene (Fig 1).

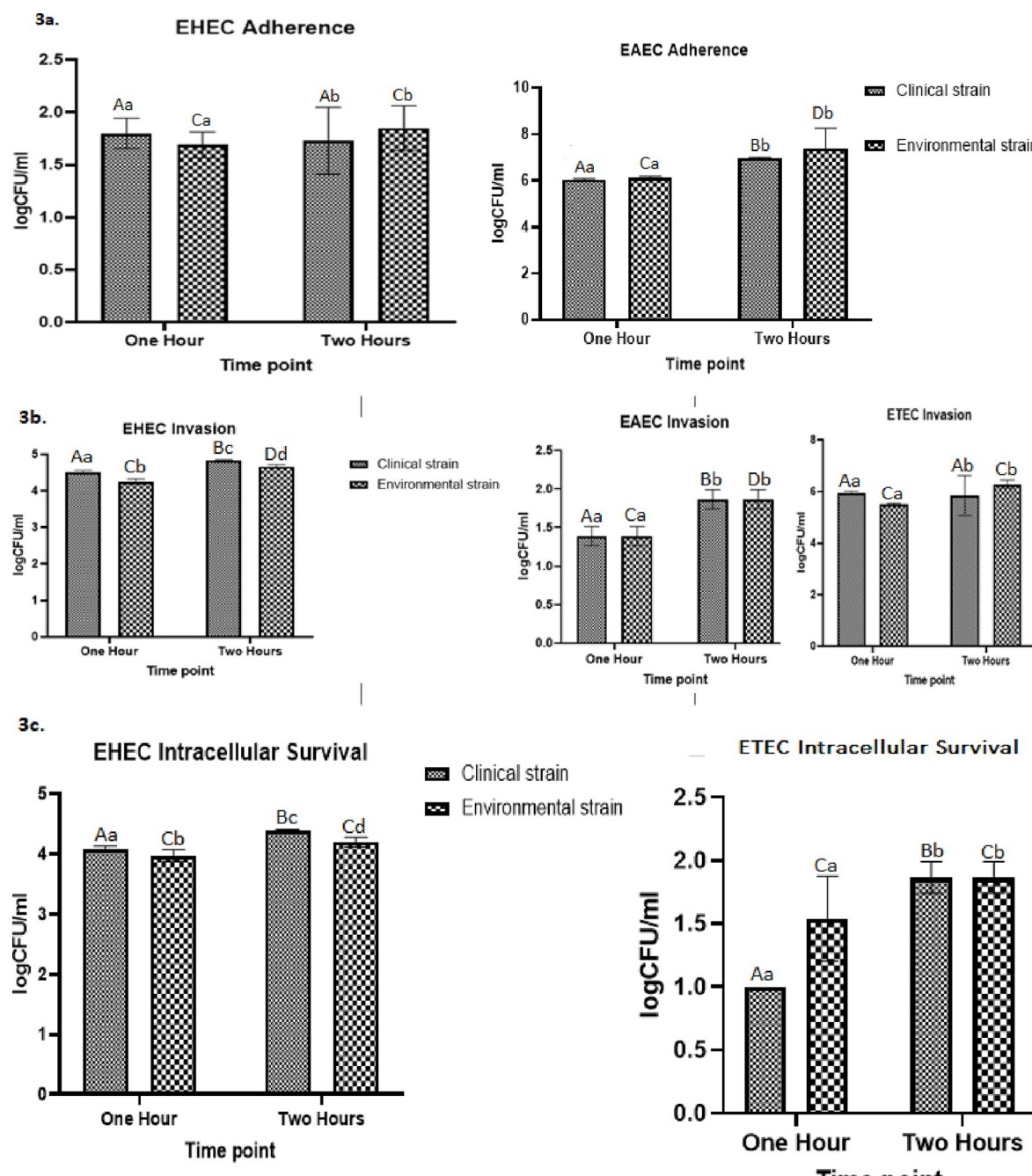

**Fig 3.** (a-c) Infectivity of Hela cells by clinical (positive control) and environmental strains of pathogenic *E. coli*. **a.** Adherence of clinical and environmental EAEC and EHEC to Hela cells. ETEC was negative for adhesion. **b.** Invasion of Hela cells by clinical and environmental ETEC, EAEC, and EHEC. **c.** Intracellular survival of clinical and environmental ETEC and EHEC in Hela cells. EAEC was negative for intracellular survival. Clinical strains refer to positive controls, and the environmental isolates to representative isolates (BHE16, ETEC; BHE59, EAEC; BHE94, EHEC) obtained in this study. At each time point, bars with similar lowercase letters indicate that adherence, invasion, or intracellular activity are not significantly different (p > 0.05) between the two strains. For each strain, bars with different upper case letters indicate that adherence, invasion, or intracellular activity are significantly different (p < 0.05) between the two time points.

These findings are similar to work done in Iran on the prevalence of enterotoxigenic *E. coli* strains in three (Imam, Mohammad Kermanshahi, and Farabi) hospital wastewater treatment plants. The study reported that from a total of 31 *E. coli* isolates, 8 pathogenic *E. coli* strains were isolated, and from the 8 pathogenic *E. coli*, a total of 25.8% of all *E. coli* strains were ETEC and harboured only the lt virulence gene [27]. In another study, done in South Africa in 2010, on the occurrence of pathogenic *E. coli* in South African wastewater, pathogenic *E. coli* were evaluated in 6 wastewater treatment plants, and 80% of the samples (from influent and effluent of wastewater treatment) were ETEC strains only harbouring the lt gene [28]. The high prevalence of the ETEC strains observed in these studies may be because these studies were done in developing countries, where ETEC is the second highest causative agent of acute diarrhoea affecting paediatrics after *Shigella* spp. [8]. However, the observed differences with our study could be due to the difference in sample size and in, the primers used in the respective studies, and geographic differences. With regards to investigating the prevalence of *E. coli* pathotypes in environmental samples, fewer studies have been done compared to clinical studies. However, ETEC prevalence in wastewater may mirror the clinical prevalence of ETEC since ETEC is associated with most hospitalised *E. coli*-associated diarrhoea [12].

EAEC, with 2 isolates (2.13%), was observed at a very low frequency as compared to ETEC (Fig 2). However, despite being at a low frequency in this study, EAEC has been reported by other studies to occur at higher frequencies in water environments [29]. A study done in Southern Brazil on the prevalence of diarrheagenic *E. coli* carrying toxin-encoding genes isolated from children and adults reported that from 56 children and 74 adults, EAEC was the most prevalent pathotype, with 23% of the isolates testing positive for EAEC virulent genes [30].

EHEC was the least prevalent pathotype with one isolate (1.06%) (Fig 2). These results were expected as EHEC usually originates in animals, such as cattle, therefore it is usually an issue in developed countries as they have lots of petting zoos and is rarely reported in developing countries [8].

The distribution observed in this study implies that hospital wastewater serves as a reservoir for some DEC pathotypes at varied levels. The high percentage of ETEC observed in this current study might be because the sampling point included wastewater from the paediatric wing in the hospital, and ETEC has been reported to be responsible for over 25% of hospitalised childhood infections in developing countries [27]. However, the difference in the frequency of these pathotypes in the different studies may be due to differences in the detection methods, geographical distribution, demographics of the areas where the studies are being done, and a difference in the sample source.

## Pathogen host interactions

Colonization of the host epithelia by pathogenic *E. coli* is primarily subject to the ability of the bacteria to interact with host surfaces. Despite knowing the prevalence of pathogenic *E. coli* in hospital wastewater, knowledge on the infectivity of such isolates remains scarce. The infectivity status of the environmental pathotypes obtained in this study was compared to clinically isolated pathotypes. The infectivity study looked at the adherence, invasion, and intracellular survival of randomly selected pathotypes (EHEC, ETEC, and EAEC) using Hela cells. The selected isolates were assayed and compared to positive controls using a 2-way ANOVA (p = 0.05).

The most prevalent pathotype in this study (ETEC), indicated that there was no significant difference in the invasion and intracellular survival (Fig 3A and 3C) between the clinical (positive control) and the environmental strain. No adherent cells were observed on the HeLa cell

surface. However, there was a significantly higher number of *E. coli* cells that intracellularly survived in the cells after 2 hours of exposure compared to after 1 hour of exposure. There was also no significant difference in the invasion of the *E. coli* cells between the two timelines. The high level of invasion was expected since ETEC is an invasive pathotype, and these results correlate to the results observed in a study done by Rappelli et al. [31]. The increase in the number of cells that survived intracellularly with an increase in time of exposure may be attributed to the possibility of multiplication of the *E. coli* cells while inside the cells.

Results for the EAEC pathotype indicated no significant difference in the adherence and invasion (Fig 3A and 3B) between the EAEC clinical sample (positive control) and the environmental isolate. However, there was a significantly higher number of *E. coli* cells that adhered and invaded, a the Hela cells after 2 hours of exposure, compared to after 1 hour of exposure. These results were expected since this pathotype is characterised by its adherence to human cells [32]. These results also correlate to the results observed in a study conducted by Marinescu et al. [1], which indicated that 81% of *E. coli* expressed aggregative/stacked brick and localised adherence patterns, meaning that these strains were either EAEC (aggregative/ stacked brick adherence patterns) or EPEC (localised adherence patterns). Intracellular survival was not observed in this pathotype. This is because none of the cells were able to proliferate and survive inside the HeLa cells. The increase in the number of the cells that adhered to the cells with an increase in the time of exposure is because more bacteria managed to adhere, and multiply during that period.

Results for the EHEC pathotypes indicated no significant difference in the adherence of the EHEC clinical (positive control) and environmental isolate (Fig 3A). However, there was no significant increase in the number of *E. coli* cells that adhered to the Hela cells after 2 hours of exposure, as compared to after 1 hour of exposure. The number of EHEC that adhered to the cell surface was considerably lower than that observed for EAEC suggesting that EHEC are a less adherent pathotype. The results of this study suggest that the environmentally isolated strains remain infective even after release into the environment. The differences in the degree of infectivity as time increases are in agreement with the results observed by Vazquez-Juarez et al. [33]. The invasion of the EHEC strains is attributed to the presence of the *stx* gene, which is a gene coding for Shiga toxins or verotoxins produced by the EHEC strains, while the adhesion of the EHEC to the HeLa cells may be attributed to the presence of the *eaeA* gene which is responsible for coding for adhesins. The high level of intracellular survival observed in EHEC is attributed to the fact that EHEC is an invasive pathotype [8]. The increase in the number of cells that survived intracellularly with an increase in time of exposure may be attributed to the possibility of multiplication of the *E. coli* cells while inside the cells.

One of the limitations of the present study was the sample size of the study and the length of the sampling period, making this a "snapshot" of the true picture of the prevalence of pathogenic *E. coli* in hospital wastewater. Since there are limited studies on the surveillance of pathogenic bacteria in the environment in Zimbabwe, there is a need for broader monitoring in different regions of the country to get a better and more accurate assessment of the prevalence of pathogenic *E. coli* in the environment.

## Conclusion

Analysis of the hospital wastewater with molecular methods revealed that the hospital is a hotspot for pathogenic *E. coli* such as ETEC, and the infectivity study revealed that the environmentally isolated pathotypes maintained their ability to colonise and infect mammalian cells through adherence and invasion of the cells even after release into the environment. The infectivity study showed that these environmentally isolated pathotypes are as infective as the

clinical pathotypes, making them a potential public health concern. This suggests that there is a possibility of the transfer of these pathogens from hospital wastewater into the community through farming and drinking water contamination if the water is not treated properly. Due to this, the wastewater of the hospitals needs to be rehabilitated to an optimal level of health with suitable treatment before release into the environment.

## Supporting information

**S1 Table. *E. coli* primers and reference strains used in PCR reactions.**
(DOCX)

**S1 Fig.** Sample collection was done from site F, the intersection point of A (Children's ward sewer), B (Intensive Care Unit sewer), C (Burns ward sewer), D (Casualty sewer), and E (Female ward sewer).
(DOCX)

**S2 Fig. Representative amplicons obtained by PCR for isolates tested for the lt gene, with the expected size of 708bp.**
(DOCX)

**S3 Fig. Amplicons obtained by PCR for isolates tested for eagg gene with the expected size of 194bp.**
(DOCX)

**S4 Fig. Amplicons obtained by PCR for isolates tested for the eaeA and stx gene, with the expected size of 248bp and 478bp.**
(DOCX)

**S5 Fig. Amplicons obtained by PCR for isolates tested for the eaeA with the expected size of 248bp.**
(DOCX)

## Acknowledgments

Mpilo Medical School Physiology laboratory staff for assisting us with the cell culturing section of this project. Vaccine-Preventable Diseases Respiratory and Meningeal Pathogens Research Unit (RMPRU) (South Africa) for providing us with the HeLa cells and the cell culturing reagents used in this project.

## Author Contributions

**Conceptualization:** Joshua Mbanga, Wilhem F. Oosthuysen.

**Data curation:** Joshua Mbanga, Wilhem F. Oosthuysen.

**Formal analysis:** Joshua Mbanga, Wilhem F. Oosthuysen.

**Investigation:** Nokukhanya P. Kodzai.

**Methodology:** Joshua Mbanga, Nokukhanya P. Kodzai.

**Supervision:** Joshua Mbanga, Wilhem F. Oosthuysen.

**Writing – review & editing:** Joshua Mbanga, Wilhem F. Oosthuysen.

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
