## [Decision Letter · Decision Letter 0]

15 Nov 2022

PONE-D-22-26831Antibiotic resistance, pathotypes and pathogen-host interactions in Escherichia coli from hospital wastewater in Bulawayo, ZimbabwePLOS ONE

Dear Dr. Mbanga,

Thank you for submitting your manuscript to PLOS ONE. After careful consideration, we feel that it has merit but does not fully meet PLOS ONE’s publication criteria as it currently stands. Therefore, we invite you to submit a revised version of the manuscript that addresses the points raised during the review process.

We look forward to receiving your revised manuscript.

Kind regards,

Sherin Reda Rouby, PhD

Academic Editor

PLOS ONE

Journal Requirements:

"This work is based on the research supported by the National University of Science and Technology (NUST) Research Board grant # RB17/43 and the kind support of the Vaccine-Preventable Diseases Respiratory and Meningeal Pathogens Research Unit (RMPRU), South Africa."

4. Please include your tables as part of your main manuscript and remove the individual files. Please note that supplementary tables (should remain/ be uploaded) as separate "supporting information" files.

Additional Editor Comments:

The paper need some major revisions before its fully acceptance

The paper need grammatical corrections. Please do it using a native English speaker.

Reviewers' comments:

Reviewer's Responses to Questions

**Comments to the Author**

1. Is the manuscript technically sound, and do the data support the conclusions?

Reviewer #1: Yes

Reviewer #2: Yes

2. Has the statistical analysis been performed appropriately and rigorously? 

Reviewer #1: Yes

Reviewer #2: No

3. Have the authors made all data underlying the findings in their manuscript fully available?

Reviewer #1: Yes

Reviewer #2: No

4. Is the manuscript presented in an intelligible fashion and written in standard English?

Reviewer #1: Yes

Reviewer #2: Yes

5. Review Comments to the Author

Reviewer #1: COMMENTS TO AUTHORS:

Manuscript Number: PONE-D-22-26831

Article Type: Research Article

Full Title: Antibiotic resistance, pathotypes and pathogen-host interactions in Escherichia coli

from hospital wastewater in Bulawayo, Zimbabwe

Short Title: Pathotypes and pathogen-host interactions in Escherichia coli from hospital

wastewater

Keywords: Antibiotic Resistance; Hospital wastewater; Zimbabwe; Escherichia coli; HeLa cells

Well, this is an original research article about the antibiotic resistance, pathotypes and pathogen-host interactions in Escherichia coli from hospital wastewater in Bulawayo, Zimbabwe. This study aimed to characterise E. coli strains, isolated from hospital wastewater effluent in Bulawayo, Zimbabwe, using both molecular and cytological approaches. In this survey, wastewater samples were aseptically collected from the sewerage mains of a major public referral hospital in Bulawayo province weekly for one month. A total of 94 isolates were isolated and confirmed as E. coli, through biotyping and PCR targeting the uidA housekeeping gene. A total of 7 genes (eagg, eaeA, stx, flicH7, ipaH, lt, and st genes) coding for virulence in diarrheagenic E. coli were targeted. Antibiotic susceptibility of E. coli was determined against a panel of 12 antibiotics through the disk diffusion assay. The infectivity status of the observed pathotypes was investigated using HeLa cells through the adherence, invasion, and intracellular assay. None of the 94 isolates tested positive for the ipaH and flicH7genes. However, 48 (53.3%) isolates were enterotoxigenic E. coli (ETEC) (lt gene positive), 2 (2.13 %) isolates were enteroaggregative E. coli (EAEC) (eagg gene), and 1 (1.06%) isolate was enterohaemorrhagic E. coli (EHEC) (stx and eaeA). A high level of sensitivity was observed in E. coli against ertapenem (98.9%), and azithromycin (75.5%). The highest resistance was against ampicillin (92.6%) and sulphamethoxazole - trimethoprim (90.4%). Seventy-nine (84%) E. coli isolates exhibited multi-drug resistance. The infectivity study results indicated that environmentally isolated pathotypes were as infective as the clinically isolated pathotypes for all three parameters. No adherent cells were observed using ETEC and no cells were observed in the intracellular survival assay using EAEC. This study revealed that hospital wastewater is a hotspot for pathogenic E. coli and, that the environmentally isolated pathotypes maintained their ability to colonise and infect mammalian cells. This is a public health concern.

The study was well-designed and interesting. However, some major revisions are required before its fully acceptance. Thus, please address my following comments one by one and re-submit the corrected manuscript.

Comments to authors:

1-Some parts of the text need grammatical and topographical corrections. Please do it using a native English speaker.

2-How many samples were collected?

3-How the sample size was measured?

4-In the method section, there was no information about the type of collected samples (liquid or solid wastewater), method of sampling, number of samples, and other information.

5-The authors utilized only the Gram staining and oxidase test to approve E. coli isolates. It is not enough to carefully confirm the E. coli isolates.

6-Why did the authors use uidA (β-D glucuronidase) gene for E. coli identification? Why they didn’t use 16SrRNA?

7-Please completely determined the method used for DNA extraction.

8-How the quality and quantity of extracted DNA samples were determined?

9-Are you sure that E. coli ATCC® 25922 is positive for the uidA gene?

10-What about the negative control in the first PCR?

11-Please add positive and negative controls for the pathotyping of E. coli strains.

12-Please add the findings of disk diffusion for your applied control E. coli ATCC 25922.

13-In my opinion, both supplementary files (table and figure) should add to the text of the paper not as supplementary.

14-In the method used for pathotyping, the pathotypes assayed for included EHEC, EPEC, EIEC, EAEC, and ETEC were identified only by PCR. However, it was not enough. Because the authors determined the EHEC/EPEC strains as those that contained eaeA gene (and stx and fliCH7 for the EHEC strains (However, how do they differentiate from EHEC and EPEC?)), EAEC as those contained the eagg gene, EIEC gene as those contained ipaH gene, ETEC those contained st and lt genes (it is not determined that both of them or only one of them?). In my opinion, the method should do with more accurate details. It is also important to add controls.

15-Please add the PCR figure of genes.

16-Which bacterial concentration was used in the disk diffusion?

17-Please determine the antibiotic resistance pattern based on the pathotype of E. coli strains, too.

18-As Chloramphenicol is a forbidden antibiotic. Why 68.1% of isolates harbored resistance against it?

19-Please add more details about the exact meaning of week 1, week 2 , …

20-Please add the study limitation.

21-Please add more interpretations about the findings of the study in the discussion section.

Cheers

Dr. Farhad Safarpoor Dehkordi

Reviewer #2: The authors describe from a short-term (1 month) study, E.coli strains isolated from hospital wastewater effluent using spread plate method on EMB agar following a serial dilution. They used conventional PCR targeting uidA gene for confirmation of the strains. The confirmed strains were subsequently pathotyped by PCR using primers targeting 7 virulence genes that categorize the isolates into 5 subgroups of DEC. A total of 94 confirmed E.coli isolates were screened for antibiotic susceptibility using disk diffusion assay. Only three isolates ETEC (BHE16), EAEC (BHE59), and EHEC (BHE94) were selected for adherence and invasion assay using HeLa cells.

Overall, the manuscript is well written and informative. I have comments and suggestions below to improve the manuscript

Comments:

Limitation in sample size…, why is only one of each pathotype selected for the adherence and invasion assays? I suggest it to be done in triplicate where possible, that will add more validation to support the conclusions.

Results are very brief, mostly just a few sentences followed by Table or Figure. I suggest the authors to combine the results and discussion, especially that they still reference figures in the discussion section.

The authors included the statistical analysis in the methods, but interrogation is lacking in the text, statistically significant results must be clearly highlighted within text and in the figures/tables

Abstract

Statement of public health concern should be removed from the last line and revised to fit as first sentence in the abstract, followed by the aims of the study.

Introduction

Line 63-64: remove “acting as a hotspot for” and replace with “facilitating”. Use ARGs acronym as already defined in line 58.

Line 77: Predatory bacteria; is defined as bacteria the eat other bacteria, I do not think it was used here in that context. Consider a most suitable term

Sample collection:

Not clear what the duplicate samples were collected for, were they all analyzed independently?

Line 95: Supplementary Figure S1, not sure what is being described in the figure? Is this the original photo, maybe a diagram will depict better

The figure shows different sites, A-F, the results will be more interesting if the samples were collected from each site?

Isolation and confirmation of E. coli:

Authors mention that the effluent sample was serially diluted from -1 to -10, 100uL was spread-plated for only -1 to -5, why was -6 up to -10 was not plated or the point of dilution upto -10

Line 114: Is there a reference for the standard heat lysis protocol? If not briefly describe the protocol.

adherence and invasion assay

line 153-154: specify the strains of positive controls used, were they ATCC or obtained elsewhere. Are these the ones mentioned as clinical strains in Figure 3, 4 and 5?

Figures 3-5 should also be combined into 1 figure, Figure 3 to be 3A, figure 4 to be 3B, and figure 5 to be figure 3C, and all graphs should use the same scale (e.g. 0.0 logcfu/mL to 10 logcfu/mL for adherence, invasion, and 0.0 to 5.0 logcfy/mL for intracellular survival ) for better comparison among the different pathotypes in three different assays

Why is ETEC (BHE16) not tested for adherence?

Why is EAEC(BHE95) not tested for intracellular survival?

Results and discussion

Line 197, 252, 257 : incorporate n=…, to clearly indicate the number of isolates

Line 264, 298 name of the genes and strains should be in italics

6. PLOS authors have the option to publish the peer review history of their article (what does this mean?). If published, this will include your full peer review and any attached files.

Reviewer #1: **Yes: **Dr. Farhad Safarpoor Dehkordi

Reviewer #2: No

---

## [Author Response · Author response to Decision Letter 0]

22 Dec 2022

RE: Rebuttal letter

Title: Antibiotic resistance, pathotypes, and pathogen-host interactions in Escherichia coli from hospital wastewater in Bulawayo, Zimbabwe. PONE-D-22-26831.

Please find below a Table in response to the comments of both reviewers. The changes to the manuscript are indicated in the text by tracked changes (reviewer 1) and in tracked changes and yellow highlight (reviewer 2).

Reviewer’s comments Response/Rebuttal

Editor’s comments

Please ensure that your manuscript meets PLOS ONE's style requirements, including those for file naming We have made the necessary amendments to meet PLOS ONE style requirements

Please include your amended statements within your cover letter; we will change the online submission form on your behalf. The funding information has been removed from the Acknowledgement section. We have included the amended funding statement in the cover letter. 

Your ethics statement should only appear in the Methods section of your manuscript. If your ethics statement is written in any section besides the Methods, please move it to the Methods section and delete it from any other section. Please ensure that your ethics statement is included in your manuscript, as the ethics statement entered into the online submission form will not be published alongside your manuscript. The Ethics statement has been moved to the Methods and materials section.

Please include your tables as part of your main manuscript and remove the individual files. Please note that supplementary tables (should remain/ be uploaded) as separate "supporting information" files. The tables have been included in the main manuscript.

Reviewer 1

Some parts of the text need grammatical and topographical corrections. Please do it using a native English speaker. We used a language editor Grammarly to edit the manuscript & sought additional assistance to improve the grammar 

How many samples were collected? Four wastewater samples were collected (duplicate samples per week for a month). This is mentioned in line 108.

In the method section, there was no information about the type of collected samples (liquid or solid wastewater), method of sampling, number of samples, and other information The type of collected sample is mentioned. line 105 ” The sampled sewerage mains receives liquid waste…” 

Method of sampling has been added – line 108

Number of samples – line 108

The authors utilized only the Gram staining and oxidase test to approve E. coli isolates. It is not enough to carefully confirm the E. coli isolates The selective media Eosin Methylene blue was used for isolation, and confirmation was primarily based on the PCR. More biochemical tests could have been necessary had we not done molecular confirmation.

Why did the authors use uidA (β-D glucuronidase) gene for E. coli identification? Why they didn’t use 16SrRNA? The uidA gene sequence is unique to E. coli (Bej et al., 1991, Appl Environ Microbiol 57:1013–1017). This means it can be routinely used for the identification of E. coli species {Green et al. 1991, https://doi.org/10.1016/0167-7012(91)90046-S } {Maheux et al. 2009, Water Res 43:3019–3028}, {Janezic et al. 2013, Open Microbiol. J. 7:9–19.}, { Naganandhini et al. 2015, 10.1371/journal.pone.0130038}, {Mbanga et al., 2021, 10.1089/mdr.2020.0380}

- I6SrRNA works for molecular identification if it is married with PCR amplicon sequencing.

Please completely determined the method used for DNA extraction. Heat lysis protocol is the boiling method a reference has been added. Line 133.

How the quality and quantity of extracted DNA samples were determined? The quality and quantity of DNA were determined using gel electrophoresis and a molecular weight marker. This has been included in the manuscript. Lines 133 – 139.

Are you sure that E. coli ATCC® 25922 is positive for the uidA gene? Yes. Very positive. As previously reported in many papers. (Titilawo et al., 2015, https://doi.org/10.1186/s12866-015-0540-3), { Naganandhini et al. 2015, 10.1371/journal.pone.0130038}, {Mbanga et al., 2021, 10.1089/mdr.2020.0380}, 

What about the negative control in the first PCR? The negative control is mentioned in line 145

Please add positive and negative controls for the pathotyping of E. coli strains. The positive controls are referred to as reference strains in line 152. To ensure there is no ambiguity we have added positive controls to the sentence in lines 152 – 153.

The negative control is mentioned in line 160.

Please add the findings of disk diffusion for your applied control E. coli ATCC 25922. We feel it is unnecessary to include this result in the main manuscript as the results of the disc diffusion assay for this control strain are well known. The isolate is susceptible to most antibiotics. Should the reviewer insist on seeing these results, they can be availed via email.

In my opinion, both supplementary files (Table and figure) should add to the text of the paper not as supplementary As suggested, we have included supplementary table S2 in the main text as Table 1. We, however, do not see the value of including supplementary Table S1 in the main text as the Table was previously published as supplementary material (Mbanga et al., 2020, https://doi.org/10.1186/s12866-020-02036-7 ). S1 Fig. is a picture of the wastewater collection site (drainage) in the main sewer of the hospital and is hardly referred to in the main text. Hence, we believe it is appropriately placed in the supplementary materials section.

-In the method used for pathotyping, the pathotypes assayed for included EHEC, EPEC, EIEC, EAEC, and ETEC were identified only by PCR. However, it was not enough. Because the authors determined the EHEC/EPEC strains as those that contained eaeA gene (and stx and fliCH7 for the EHEC strains (However, how do they differentiate from EHEC and EPEC?)), EAEC as those contained the eagg gene, EIEC gene as those contained ipaH gene, ETEC those contained st and lt genes (it is not determined that both of them or only one of them?). In my opinion, the method should do with more accurate details. It is also important to add controls. The method used to pathotype E. coli in our study is a standard protocol that has been reported in several studies. We list a few below.

• https://doi.org/10.1016/j.scitotenv.2016.05.155. 

• https://doi.org/10.1111/1469-0691.12646. 

• https://doi.org/10.1186/s13099-019-0290-0. 

• Omar and Barnard, 2014, World J Microbiol Biotechnol. 2014;30(10):2663–71

• Adesifoye et al. 2016, Microbiology open. 2016;5(1):143–51

We stand guided on what additional assays the reviewer feels should have been done. 

Please add the PCR figure of genes. These have been added as supplementary material. 

Which bacterial concentration was used in the disk diffusion? The antibiotic susceptibility testing and interpretation of results were done according to CLSI guidelines. This has been clarified in lines 182 -183. Bacterial concentration should be equivalent to a 0.5 MacFarland standard.

Please determine the antibiotic resistance pattern based on the pathotype of E. coli strains, too. The manuscript includes antibiotic resistance patterns based on the E. coli pathotype in Table 4.

As Chloramphenicol is a forbidden antibiotic. Why 68.1% of isolates harbored resistance against it? We need clarification on the question. Is the expectation that there should be no resistance to chloramphenicol because it is banned? Resistance mechanisms are determined by a myriad of factors, including mobile genetic elements which carry a variety of resistant determinants that work against a plethora of antibiotics. Antibiotic resistance is influenced by many factors we might not fully understand for instance, why we find antibiotic-resistant bacteria in pristine environments. 

Please add more details about the exact meaning of week 1, week 2 , … This request is not clear. We sampled for four weeks, from week 1 to week 4.

Please add the study limitation Limitations are mentioned in lines 451 – 456.

Please add more interpretations about the findings of the study in the discussion section. The discussion has been expanded as suggested. 

Reviewer 2

Limitation in sample size…, why is only one of each pathotype selected for the adherence and invasion assays? I suggest it to be done in triplicate where possible, that will add more validation to support the conclusions Thank you for your comment. Indeed using more isolates would have improved the validity of the results. However, we used one isolate from each pathotype mainly because we had only one EHEC isolate. We reasoned that our experimental isolates and the positive controls used in the study would give us reliable results. 

Results are very brief, mostly just a few sentences followed by a Table or Figure. I suggest the authors to combine the results and discussion, especially that they still reference figures in the discussion section. We have added two tables to the manuscript and expanded the results section. We hope this is sufficient for the manuscript to retain the individual results and discussion sections.

The authors included the statistical analysis in the methods, but interrogation is lacking in the text, statistically significant results must be clearly highlighted within text and in the figures/tables This has been done in the figures (Fig. 3a-c) and intext. 

Abstract

Statement of public health concern should be removed from the last line and revised to fit as first sentence in the abstract, followed by the aims of the study. The statement has been removed from the abstract entirely.

Introduction

Line 63-64: remove “acting as a hotspot for” and replace with “facilitating”. Use ARGs acronym as already defined in line 58 The suggestions have been implemented, facilitating added - line 67. ARG acronym used – line 67

Line 77: Predatory bacteria; is defined as bacteria that eat other bacteria, I do not think it was used here in that context. Consider a most suitable term The term predatory has been replaced by pathogenic – line 83

Sample collection:

Not clear what the duplicate samples were collected for, were they all analyzed independently? The duplicate samples were pooled and analysed as one composite sample. Lines 110 – 111.

Line 95: Supplementary Figure S1, not sure what is being described in the figure? Is this the original photo, maybe a diagram will depict better

The figure shows different sites, A-F, the results will be more interesting if the samples were collected from each site? Figure S1 is a photograph of the main sewer where wastewater was collected. It shows that effluent from different wards merges into one at the sampled point. We concur that the wastewater could have been sampled per respective ward, however, this was out of the scope of this study but should be considered in future studies.

Authors mention that the effluent sample was serially diluted from -1 to -10, 100uL was spread-plated for only -1 to -5, why was -6 up to -10 was not plated or the point of dilution upto -10 Thank you for this observation. We made dilutions up to -5 not -10 the mistake has been corrected. – Line 121. 

Line 114: Is there a reference for the standard heat lysis protocol? If not briefly describe the protocol.

adherence and invasion assay A reference has been added – line 133

line 153-154: specify the strains of positive controls used, were they ATCC or obtained elsewhere. Are these the ones mentioned as clinical strains in Figure 3, 4, and 5? A statement specifying the positive controls/reference strains has been added. DSM isolates were used. Lines 152 -154. The same reference isolates were used for the infectivity study.

Figures 3-5 should also be combined into 1 figure, Figure 3 to be 3A, figure 4 to be 3B, and figure 5 to be figure 3C, and all graphs should use the same scale (e.g. 0.0 logcfu/mL to 10 logcfu/mL for adherence, invasion, and 0.0 to 5.0 logcfy/mL for intracellular survival ) for better comparison among the different pathotypes in three different assays Because of the differences in the results of the assayed pathotypes the software gave us the best output by determining the scale for the Y-axis. We have, however, reorganised the Figs to 3a. showing the adherence assay, 3b. showing the invasion assay and 3c. showing the intracellular survival assay for easy comparison among the different pathotypes in the three different assays. Fig.3a-c.

Why is ETEC (BHE16) not tested for adherence? The isolate was tested for adherence but yielded negative results. Lines 302-303

Why is EAEC(BHE95) not tested for intracellular survival? The isolate was tested for adherence but yielded negative results. Lines 306-307

Results and discussion

Line 197, 252, 257 : incorporate n=…, to clearly indicate the number of isolates This has been amended lines 263 – 265.

Line 264, 298 name of the genes and strains should be in italics This has been amended in the entire document.

---

## [Decision Letter · Decision Letter 1]

23 Jan 2023

PONE-D-22-26831R1Antibiotic resistance, pathotypes and pathogen-host interactions in Escherichia coli from hospital wastewater in Bulawayo, ZimbabwePLOS ONE

Dear Dr. Mbanga,

Thank you for submitting your manuscript to PLOS ONE. After careful consideration, we feel that it has merit but does not fully meet PLOS ONE’s publication criteria as it currently stands. Therefore, we invite you to submit a revised version of the manuscript that addresses the points raised during the review process.

ACADEMIC EDITOR: Authors should change the manuscript carefully according to reviewer recommendations to be suitable for publication. />==============================

We look forward to receiving your revised manuscript.

Kind regards,

Sherin Reda Rouby, PhD

Academic Editor

PLOS ONE

Reviewers' comments:

Reviewer's Responses to Questions

**Comments to the Author**

1. If the authors have adequately addressed your comments raised in a previous round of review and you feel that this manuscript is now acceptable for publication, you may indicate that here to bypass the “Comments to the Author” section, enter your conflict of interest statement in the “Confidential to Editor” section, and submit your "Accept" recommendation.

Reviewer #1: (No Response)

Reviewer #2: All comments have been addressed

2. Is the manuscript technically sound, and do the data support the conclusions?

Reviewer #1: No

Reviewer #2: Yes

3. Has the statistical analysis been performed appropriately and rigorously? 

Reviewer #1: Yes

Reviewer #2: I Don't Know

4. Have the authors made all data underlying the findings in their manuscript fully available?

Reviewer #1: Yes

Reviewer #2: Yes

5. Is the manuscript presented in an intelligible fashion and written in standard English?

Reviewer #1: No

Reviewer #2: Yes

6. Review Comments to the Author

Reviewer #1: Manuscript Number: PONE-D-22-26831R1

Manuscript Title: Antibiotic resistance, pathotypes and pathogen-host interactions in Escherichia coli from hospital wastewater in Bulawayo, Zimbabwe

Journal: Plos One

Comments to authors:

Well, this is an original research study about the Antibiotic resistance properties, pathotypes distribution and pathogen-host interactions in Escherichia coli strains isolated from hospital wastewater in Bulawayo, Zimbabwe.

Some of my comments have been addressed. However, important items should clearly identify before the acceptance of the paper.

1-The authors mentioned that Casualty department, Burns ward, Intensive Care Unit (ICU), Female ward, and Children’s ward were targeted. Is there any differences between properties your assessed or not?

2-How the sample size (A total of 4 wastewater samples were collected in duplicate) was determined?

3-Add reference for the molecular confirmation of E. coli. Add primers

4-I have a question, are all E. coli strains harbored the uidA (β-D glucuronidase) genes? In total I mean.

5-How the pathotypes were determined? We need to know more about primers and related genes. Please add its table in the text. Additionally, please clearly explained that which two or one genes correspond for each pathotypes?

6-S4 fig: In my opinion your figure is not correspond for these genes. It is because of your ladder is not 100 bp ladder. If a ladder was 100 pb, the 500 bp stair should clearly highlighted and more predominant!!!! Similar to S2, S3, S5. This is the main figure of 100bp ladder:

Pay attention to my points (unfortunately I couldn't attached file here, Check the exact 100 bp ladder here: https://international.neb.com/products/n3231-100-bp-dna-ladder#Product%20Information)

7-As far as we know, the EHEC strains are mainly coming from animals. How did you interprate this?

8-Which bacterial concentration was used in disk diffusion?

9-What differences were performed to growth each pathotypes in the Muller Hinton agar?

10-I checked the CLSI version you used in this study. The mean diameter of the growth inhibition zone of azithromycin was not determined in this version.

11-Please also add the results of positive and negative controls for all disks, genes, and …

12-As I said before, remove the text and add all PCR conditions including targeted genes, sequence of primers, size of products, Thermal cycling and volumes in tables.

13-The method used for bacterial isolation using culture is not accurately determined. There were no biochemical tests. Additionally, the authors tried to measure the bacterial colonies in each millimeter of the swage samples but their results are not presented.

14-Additionally, if the authors did bacterial count, they should compared them with standard of Europe and their country and also US FDA.

Sincerely

FSD

Reviewer #2: The authors have sufficiently addressed most of the comments raised by the reviewers. Only one reservation is on the negative control. In my view the negative control in this case must be bacterial DNA that is known to be negative of the gene that is targeted. Using nuclease free water as negative only confirms that there is no contamination.

7. PLOS authors have the option to publish the peer review history of their article (what does this mean?). If published, this will include your full peer review and any attached files.

Reviewer #1: **Yes: **Dr. Farhad Safarpoor Dehkordi (FSD)

Reviewer #2: No

---

## [Author Response · Author response to Decision Letter 1]

1 Feb 2023

Dear Dr. Rouby

Academic Editor: PLOS ONE

RE: Rebuttal letter

Title: Antibiotic resistance, pathotypes, and pathogen-host interactions in Escherichia coli from hospital wastewater in Bulawayo, Zimbabwe. PONE-D-22-26831.

Please find below a Table in response to the comments of both reviewers. The changes to the manuscript are indicated in the text by tracked changes (reviewer 1) and in tracked changes and yellow highlight (reviewer 2).

Reviewer’s comments Response/Rebuttal

Reviewer 1

Well, this is an original research study about the Antibiotic resistance properties, pathotypes distribution and pathogen-host interactions in Escherichia coli strains isolated from hospital wastewater in Bulawayo, Zimbabwe. Thank you. We concur.

The authors mentioned that Casualty department, Burns ward, Intensive Care Unit (ICU), Female ward, and Children’s ward were targeted. Is there any differences between properties your assessed or not? We sampled from the sewer mains that receives sewage from the casualty department, burns ward, Intensive care unit, female ward, and Children’s wards. We did not collect samples from the sewers of each ward independently and thus could not make comparisons based on the different wards. We hope this is clear.

How the sample size (A total of 4 wastewater samples were collected in duplicate) was determined? Determination of sample size was not calculated statistically but based on referring to other similar cross sectional studies on the environment. This was notably a limitation but we believe the sample size did not mask the resultant findings revealed by the cytology part of the study. We did mention that sample size was a limitation of the study. Line 418 -419.

Add reference for the molecular confirmation of E. coli. Add primers A reference has been inserted in line 138.

Primer sequences have been added. lines 140 & 141.

I have a question, are all E. coli strains harbored the uidA (β-D glucuronidase) genes? In total I mean. Typically upwards of 95% of E. coli possess the β-D glucuronidase gene hence its wide use for the detection of the microorganism. The uidA gene sequence is unique to E. coli (Bej et al., 1991, Appl Environ Microbiol 57:1013–1017). This means it can be routinely used for the identification of E. coli species {Green et al. 1991, https://doi.org/10.1016/0167-7012(91)90046-S } {Maheux et al. 2009, Water Res 43:3019–3028}, {Janezic et al. 2013, Open Microbiol. J. 7:9–19.}, { Naganandhini et al. 2015, 10.1371/journal.pone.0130038}, {Mbanga et al., 2021, 10.1089/mdr.2020.0380}

How the pathotypes were determined? We need to know more about primers and related genes. Please add its table in the text. Additionally, please clearly explained that which two or one genes correspond for each pathotypes? Pathotyping of E. coli is detailed in the manuscript. Lines 150 – 168. Primers and positive controls are provided in the supplementary materials SI Table. Line 154. The genes and their corresponding pathotypes are provided in the S1 Table. 

S4 fig: In my opinion your figure is not correspond for these genes. It is because of your ladder is not 100 bp ladder. If a ladder was 100 pb, the 500 bp stair should clearly highlighted and more predominant!!!! Similar to S2, S3, S5. This is the main figure of 100bp ladder: Pay attention to my points (unfortunately I couldn't attached file here, Check the exact 100 bp ladder here: https://international.neb.com/products/n3231-100-bp-dna-ladder#Product%20Information) We thank the reviewer for the sharp observation. We have cross-checked and can confirm that the correct ladder we used was the Invitrogen 1kb plus ladder from Thermo Fisher Scientific whose brightest band is the 1.5kb band (https://assets.fishersci.com/TFS-Assets/LSG/manuals/1Kb_Plus_DNA_ladder_man.pdf). The error has been corrected in the manuscript and Figs. The gene sizes are correct.

As far as we know, the EHEC strains are mainly coming from animals. How did you interprate this? The EHEC group has been recognized as the universal cause of serious human gastrointestinal diseases Griffin et al. 1991 Epidemiol Rev. 1991;13:60–98. We have previously reported EHEC in the water environment Mbanga et al., 2020 (BMC Microbiology 20:346. Although it is predominantly found in animals it does occur in clinical settings and in the environment. As our study was in a clinical setting we can only assume that there were patients infected with EHEC during our sampling period.

Which bacterial concentration was used in disk diffusion? Bacterial concentration was equivalent to a 0.5 MacFarland standard. This provides an optical density comparable to the density of a bacterial suspension 1.5x 10^8 colony-forming units (CFU/ml). This has been added. Line 174 – 175.

What differences were performed to growth each pathotypes in the Muller Hinton agar? We, unfortunately, do not understand this question. Is the reviewer asking if there were differences observed on MUELLER Hinton depending on the pathotype?

I checked the CLSI version you used in this study. The mean diameter of the growth inhibition zone of azithromycin was not determined in this version. We have added a reference to an older version of the CLSI guidelines that gave zone diametres for Enterobacteriaceae. Line 182.

Please also add the results of positive and negative controls for all disks, genes, and… We believe this is adequately done throughout the manuscript. 

As I said before, remove the text and add all PCR conditions including targeted genes, sequence of primers, size of products, Thermal cycling and volumes in tables. We thank the reviewer once again for this suggestion. We, however, still believe our PCR conditions and volumes are amply described in text format. We have tabulated primer sequences and the expected amplicon sizes in S1 Table.

The method used for bacterial isolation using culture is not accurately determined. There were no biochemical tests. Additionally, the authors tried to measure the bacterial colonies in each millimeter of the swage samples but their results are not presented. We used the Eosin methylene blue agar to culture the E. coli. Presumptive E. coli colonies with a metallic green sheen were subjected to the Gram stain and oxidase test (Biochemical tests). Lines 120 - 127 

We did not measure or try to measure bacterial colonies in each millimetre of sewage samples. We are confused by what the reviewer is referring to here. We report that we randomly picked colonies from each sewage sample such that we had 10 representative isolates for each sample. This was not an attempt to ‘measure’or count the colonies. Lines 123 -126.

Additionally, if the authors did bacterial count, they should compared them with standard of Europe and their country and also US FDA. We did not do any bacterial counts.

Reviewer 2

The authors have sufficiently addressed most of the comments raised by the reviewers. Only one reservation is on the negative control. In my view the negative control in this case must be bacterial DNA that is known to be negative of the gene that is targeted. Using nuclease free water as negative only confirms that there is no contamination. Thank you for your comment. Indeed that is correct. The nuclease-free water does indicate that there was no contamination during the PCR and that the positive results were most likely due to the DNA template used. We will make sure to add DNA from a known negative in future studies.

---

## [Editor Report · Decision Letter 2]

13 Feb 2023

Antibiotic resistance, pathotypes and pathogen-host interactions in Escherichia coli from hospital wastewater in Bulawayo, Zimbabwe

PONE-D-22-26831R2

Dear Dr. Joshua Mbanga,

We are pleased to inform you that your manuscript has been judged scientifically suitable for publication and will be formally accepted for publication once it meets all outstanding technical requirements.

Kind regards,

Sherin Reda Rouby, PhD

Academic Editor

PLOS ONE
---

## [Editor Report · Acceptance letter]

17 Feb 2023

PONE-D-22-26831R2 

Antibiotic resistance, pathotypes, and pathogen-host interactions in *Escherichia coli* from hospital wastewater in Bulawayo, Zimbabwe 

Dear Dr. Mbanga:

I'm pleased to inform you that your manuscript has been deemed suitable for publication in PLOS ONE. Congratulations! Your manuscript is now with our production department. 

Kind regards, 

on behalf of

Professor Sherin Reda Rouby 

Academic Editor

PLOS ONE